# Sodium and Potassium Excretion of Schoolchildren and Relationship with Their Family Excretion in China

**DOI:** 10.3390/nu13082864

**Published:** 2021-08-20

**Authors:** Yuan Li, Yuewen Sun, Xian Li, Le Dong, Fengzhuo Cheng, Rong Luo, Changqiong Wang, Jing Song, Feng J. He, Graham A. MacGregor, Puhong Zhang

**Affiliations:** 1The George Institute for Global Health at Peking University Health Science Centre, Beijing 100600, China; yli@georgeinstitute.org.cn (Y.L.); ysun2@georgeinstitute.org.cn (Y.S.); lxian@georgeinstitute.org.cn (X.L.); ldong@georgeinstitute.org.cn (L.D.); fcheng@georgeinstitute.org.cn (F.C.); rluo@georgeinstitute.org.cn (R.L.); 2Faculty of Medicine, University of New South Wales, Sydney, NSW 2052, Australia; 3Wolfson Institute of Preventive Medicine, Barts and The London School of Medicine & Dentistry, Queen Mary University of London, London EC1M 6BQ, UK; changqiong.wang@qmul.ac.uk (C.W.); jing.song@qmul.ac.uk (J.S.); f.he@qmul.ac.uk (F.J.H.); g.macgregor@qmul.ac.uk (G.A.M.)

**Keywords:** sodium, salt, potassium, urine, family resemblance, children

## Abstract

This cross-sectional study aimed to assess 24-h urinary sodium and potassium excretion in children and the relationships with their family excretion. Using the baseline data of a randomized trial conducted in three cities of China in 2018, a total of 590 children (mean age 8.6 ± 0.4 years) and 1180 adults (mean age 45.8 ± 12.9 years) from 592 families had one or two complete 24-h urine collections. The average sodium, potassium excretion and sodium-to-potassium molar ratio of children were 2180.9 ± 787.1 mg/d (equivalent to 5.5 ± 2.0 g/d of salt), 955.6 ± 310.1 mg/d and 4.2 ± 1.7 respectively, with 77.1% of the participants exceeding the sodium recommendation and 100% below the proposed potassium intake. In mixed models adjusting for confounders, every 1 mg/d increase in sodium excretion of adult family members was associated with a 0.11 mg/d (95% CI: 0.06 to 0.16, *p* < 0.0001) increase in sodium excretion of children. The family-child regression coefficient corresponds to 0.20 mg/d (95% CI: 0.15 to 0.26, *p* < 0.0001) per 1 mg/d in potassium and to 0.36 (95% CI: 0.26 to 0.45, *p* < 0.0001) in sodium-to-potassium molar ratio. Children in China are consuming too much sodium and significantly inadequate potassium. The sodium, potassium excretion and sodium-to-potassium ratio of children are associated with their family excretions in small to moderate extent. Efforts are warranted to support salt reduction and potassium enhancement in children through comprehensive strategies engaging with families, schools and food environments.

## 1. Introduction

High blood pressure increases the risk of cardiovascular disease, which is the leading cause of non-communicable diseases, responsible for 31% of all deaths worldwide [1]. Excess dietary sodium intake is an important risk factor of increased blood pressure, while high potassium intake is related to lower blood pressure [2,3]. Our previous study has shown a typical high sodium and low potassium intake pattern in Chinese adults, with sodium positively and potassium inversely related to both systolic and diastolic blood pressure [4]. A recent meta-analysis including the 24-h urine collections of about 800 children in China reported a similar intake pattern, with approximately 5 g/d salt intake for children aged 3–6 years, nearly 9 g/d for those aged 6–16 years, and less than 1 g/d potassium intake in all the children [5]. Whereas the upper limits of recommended sodium intake is 3 g/d for children aged 4–6, 4 g/d for children aged 7–10 and 6 g/d for those aged 11–16 according to Chinese Dietary Guidelines, and the proposed intake of potassium for preventing non-communicable chronic diseases is 2.8 g/d for children aged 7–10 according to Chinese Dietary Reference Intakes [6,7].

Previous reports showing high sodium and low potassium intake among children deserve further investigation as the dietary intake of children would be largely affected by their upbringing environments. It was also reported that the sodium and potassium intake of children is not only correlated with their families, but also related to the children’s blood pressure, which may have a long-lasting effect on the health of these children [8,9,10]. However, many of these studies seldom used 24-h urine collection for measurement of sodium and potassium intake, which is regarded as the most accurate method for sodium and potassium assessment. As this method is time-consuming and requires high compliance of participants [11], the 24-h urine data of children was usually less available than that of adults. Using data from a study with two consecutive 24-h urine collections for both children and their adult family members, this study aims to measure the levels of sodium and potassium intake of children and to investigate their correlations within the family so as to provide references for improving the nutrient intake of children in China.

## 2. Materials and Methods

The present cross-sectional study used the baseline data of an app-based education program to reduce salt intake in schoolchildren and their families in China (registered on the Chinese Clinical Trial Registry, ChiCTR1800017553). The details of this program have been described elsewhere [12].

### 2.1. Participants

The survey was conducted in 54 primary schools from 3 cities, i.e., Shijiazhuang in northern (Hebei province), Luzhou in central (Sichuan province) and Yueyang in southern (Hunan province) China, with 18 schools from each city. From each school, we randomly selected 11 children and 22 adult family members (i.e., 1 child and 2 adults per family) from one recruited class in grade 3 for the survey. The participants of this study were family members living together or at least ate homemade meals most of the times. The inclusion criteria were (1) children and their adult family members ate homemade meals at least four times per week; (2) one of the adults in the family had a mobile device with access to the Internet; (3) if more than 2 adults in one family agreed to take part in the assessment, we selected two in the order of grandparents, parents, uncles and aunts; and (4) participants had been local residents for over 6 months. Individuals who could not or refused to collect 24-h urine, were excluded from the assessment. Grandparents were prior to parents in the selection order because of the two considerations: (1) to ensure the proportion of grandparents in the sample; otherwise, most adult participants would be parents as they are the most available family members for children; (2) the objective of the trial is to examine the impact of sodium reduction on blood pressure, which would be more meaningful for grandparents who have higher levels of blood pressure than parents.

### 2.2. Data Collection

Families including children and adults were appointed to go to school at the same time for data collection. All participants were asked to conduct two consecutive 24-h urine collections under the instructions of trained research staff, usually on one weekday and one weekend. All urine collection equipment (including containers and collection aids such as carrier bags) were provided to the participants at the investigation venue (in schools). The start time of the 24-h urine collection was recorded by the researchers after the participants emptied their bladders at the investigation venue, and the finish time was recorded at the same time on the second day after the participant came back and passed their last urine in the container. Another set of urine collection equipment was provided to the participants for their second collection upon completion of their first 24-h urine collection. If one or more urine voids were missed or spillage occurred, participants would be asked to complete another 24-h urine collection. Urine weight was measured by standardized validated scales accurate to 0.1 g. Urine samples were then extracted with 2-mL sampling tubes from the evenly mixed 24-h urine of each participant and transported in full cold train (−20 °C) to a centralized laboratory within one month. Urinary sodium, potassium and creatinine concentrations were measured in the centralized laboratory by ion-selective electrode method (sodium and potassium) and enzymatic method (creatinine). Body height and weight were also measured by validated scales at the investigation venue.

A specially designed mobile device-based electronic data capture system was developed for data collection and data management. This system not only provided timely instructions at each step of the urine collection procedure, but also maintained a convenient critical data record for quality checks, leading to a great increase in compliance of the participants. The data collection lasted 3 months from September to November of 2018, with one month for each city.

### 2.3. Definition of Variables

The 24-h urine samples were measured for volume, sodium, potassium and creatinine. The conversion and definition of these indicators can be found elsewhere [4]. For children, a salt intake transferred from sodium excretion higher than 4 g/d was defined as an excess intake of salt; a potassium intake estimated from potassium excretion below 2800 mg/d (proposed intakes for preventing non-communicable chronic diseases, PI) was defined as an insufficient intake of potassium [6,7]. Urine data were defined as incomplete if the 24-h urine volume <500 mL or creatinine <4.0 mmol for women or <6.0 mmol for men, and in children urine volume <300 mL/24 h, or 24-h urinary creatinine <5th centile for boys and girls separately [13], or the urine collections <20 h or >28 h. Body mass index (BMI) was calculated as body weight (kg) divided by the square of height (m^2^).

### 2.4. Statistical Analysis

There were no missing values for the analyzed data. Mean and standard deviation (SD) or frequency and proportion (%) were used to describe the characteristics of participants where appropriate. The data for sodium and potassium excretions of the two consecutive complete 24-h urines were averaged. If one day’s urine collection was incomplete, then only the data of the complete day were used. T test was performed to compare the difference between boys and girls in sodium, potassium excretion and sodium-to-potassium ratio. Familial correlation was performed with the Pearson’s correlation coefficients between children and the average of their two adult family members. Distributions of sodium, potassium excretion and sodium-to-potassium molar ratio of the children and their adult family members were displayed in scattered plots with regression fitted line and parameter estimates from single factor regression analysis. Multivariate mixed models were further applied to investigate the relationship of sodium, potassium excretion and sodium-to-potassium ratio of children with the corresponding excretion of adult family members. Mixed models were performed with city and school nested within city as random factors and adjusted potential confounders including age, sex and body weight of children in different models. In consideration of the better goodness-of-fit of body weight in this study, body weight of children instead of BMI was finally used in models to reflect the degree of obesity for children at the stage of growth and development. The intraclass correlation coefficients (ICC) of schools for sodium, potassium excretion and sodium-to-potassium ratio of schoolchildren were calculated in mixed models with zero independent variables. Subgroup analysis of associations of sodium, potassium excretion and sodium-to-potassium ratio of children with different family members including mother, father, grandmother and grandfather were performed in separate mixed models.

All analyses were two sided and *p* < 0.05 was considered significant. SAS Enterprise Guide 8.3 was used for analyses.

## 3. Results

### 3.1. Characteristics of Participants

A total of 592 children and 1184 adults from 592 families were investigated in three cities. Excluding six incomplete urine collection for two days, the final sample size was 590 children and 1180 adults from 592 families (2 families had only the data of adults after dropping incomplete data), with 65 children and 61 adults with only one day of complete urine collection (Figure 1).

Among the 590 children (52.2% for boys), the average age was 8.6 years (SD 0.4). Of the 1180 adult family members with 46.5% being male and mean age of 45.8 years (SD 12.9), 318 (53.7%) were parent pairs, 137 (23.1%) were grandparent pairs and 137 (23.1%) were mixed with either one parent, one grandparent, or both, or with other relatives. Other characteristics are shown in Table 1.

### 3.2. Twenty-Four-Hour Urinary Sodium, Potassium Excretion and Sodium-to-Potassium Ratio of Children

Overall, the average sodium excretion of children was 2180.9 (SD 787.1) mg/d corresponding to an average salt intake of 5.5 (SD 2.0) g/d, with 6.1 (SD 2.1) g/d in boys and 4.9 (1.7) g/d in girls (boys vs. girls, *t* = 8.03, *p* < 0.0001). The average potassium excretion of children was 955.6 (SD 310.1) mg/d, with 998.1 (SD 322.5) mg/d in boys and 909.2 (SD 289.4) mg/d in girls (boys vs. girls, *t* = 3.51, *p* = 0.0005). The mean sodium-to-potassium molar ratio of children was 4.2 (SD 1.7), with 4.5 (SD 1.9) in boys and 3.9 (SD 1.5) in girls (boys vs. girls, *t* = 4.61, *p* < 0.0001). When the intakes were estimated with excretions, 455 (77.1%) children had salt intakes exceeding the upper limit of 4 g/d recommended by Chinese Dietary Guidelines, and all of the 590 children had inadequate potassium intake below 2800 mg/d, the proposed intake for preventing non-communicable chronic diseases in Chinese Dietary Reference Intakes. (Table 1).

### 3.3. Association between Sodium, Potassium Excretion and Sodium-to-Potassium Ratio of Children, and the Corresponding Intake of Adult Family Members

The correlation coefficients between children and their adult family members of sodium, potassium excretion and sodium-to-potassium molar ratio were 0.223 (95% CI: 0.145, 0.299), 0.305 (95% CI: 0.229, 0.376) and 0.289 (95% CI: 0.213, 0.361), respectively, all with *p* < 0.0001. Other familial correlations can also be found in body height (r = 0.507), body weight (r = 0.385), BMI (r = 0.304) as well as urine volume and creatinine (Table 2). The distribution of sodium excretion of the children and their adult family members showed a fitted slope of 0.16, 0.22 and 0.36 for sodium, potassium, and sodium-to-potassium ratio, respectively (Figure 2).

In mixed models, adjusting for age, sex and body weight of children, every 1 mg/d increase of sodium excretion of adult family members was associated with a 0.11 mg/d (95% CI: 0.06 to 0.16, *p* < 0.0001) increase in sodium intake of children; each 1 mg/d increase of potassium intake of adult family members was associated with a 0.20 mg/d (95% CI: 0.15 to 0.26, *p* < 0.0001) increase in potassium intake of children; and each unit of sodium-to-potassium molar ratio of adult family members was associated with a 0.36 (95% CI: 0.26 to 0.45, *p* < 0.0001) unit increase in that of children (Table 3). Associations with different family members including mother, father, grandmother and grandfather showed that mothers were significantly aggregated with children in all the models except in sodium excretion of girls (e.g., sodium of boys, β = 0.14, *p* = 0.0002; sodium of girls, β = 0.07, *p* = 0.065); fathers showed insignificant association in sodium excretion but significant association with children in potassium excretion and sodium-to-potassium ratio (e.g., sodium of boys, β = 0.04, *p* = 0.247; potassium of boys, β = 0.21, *p* < 0.0001; sodium-to-potassium of boys, β = 0.32, *p* < 0.0001). Grandmothers showed significant association with boys in sodium (β = 0.15, *p* = 0.009) and with girls in sodium-to-potassium ratio (β = 0.28, *p* = 0.002). Grandfathers showed significant associations with total children in sodium (β = 0.09, *p* = 0.013) and sodium-to-potassium ratio (β = 0.17, *p* = 0.015) but the associations became insignificant in separate models for boys and girls (Table 4). In addition, the ICC of schools were 0.042, 0.056 and 0.039 for sodium, potassium excretion and sodium-to-potassium ratio of schoolchildren, respectively.

## 4. Discussion

This study showed that the mean sodium excretion in children averagely aged 8.6 years was about 2181 mg/d (equivalent to 5.5 g/d salt), the mean potassium excretion was about 956 mg/d and the sodium-to-potassium molar ratio was 4.2, with 77.1% above the recommended levels of sodium intake and 100% below the proposed potassium intakes for this age group. This high sodium and low potassium excretion status in children was highly in accordance with the situation in adults, with over 11 g/d salt intake (93.5% above recommended intake) and under 1600 mg/d potassium intake (99.0% under recommended intake) and 5 of sodium-to-potassium ratio in our previous study in adults, indicating that excess sodium and inadequate potassium were prevalent across the whole population in China [4]. The intake pattern of children was similar to those in other countries reporting 24-h urinary excretion, with 6.1 g/d salt intake, 1833 mg/d potassium intake and 2.4 sodium-to-potassium molar ratio in children aged 9.3 in Australia [14]; an average of 7.8 g/d salt intake, 1927 mg/d potassium intake and 2.8 molar ratio in Spanish children aged 7–11 years [15]; and 5.7 g/d salt intake, 1134 mg/d potassium intake and 4.5 sodium-to-potassium ratio of children aged 9–11 years in Japan [16]. The potassium intake appeared lower in Asian countries than that of Western countries, resulting in sodium-to-potassium ratios higher than 4 in China and Japan. The high sodium and low potassium intake pattern during childhood is likely to track into adulthood and increases the potential risks of cardiovascular disease later in life [17]. More efforts are warranted to reduce salt and increase potassium intake in children. Salt reduction has been included in the goals of Healthy China 2030 Actions. Given the evidently lower potassium intake of children in China, public health measures aimed at increasing the consumption of potassium-rich foods such as fruits, vegetables and beans, should take place simultaneously.

Our results found generally significant familial correlations between sodium, potassium excretion and sodium-to-potassium ratio of children and their adult family members. Subgroup analysis showed heterogeneity in terms of relationships with different family members, with almost all nutrients aggregated with mothers, sodium and sodium-to-potassium ratio aggregated with grandmothers, potassium and sodium-to-potassium ratio with fathers and insignificant aggregation with grandfathers in boys and girls. The heterogeneity could be partly resulted from the small sample size in subgroup analysis. For example, grandfathers showed significant associations with total children in sodium and sodium-to-potassium ratio, but the associations became insignificant in separate models for boys and girls. Some other studies on parent–child concordance of dietary choices reported small-to-moderate association with correlation coefficients no higher than 0.2 [8]. Service et al. reported that a 1 g/day increase in the mother’s salt intake was associated with a 0.20 g/day increase in the child’s salt intake and no association was found in potassium excretion [18]. However, some studies showed strong resemblance. Koyama et al. reported a mother–child correlation of 0.6 in daily total sodium intake and 0.41 in potassium intake in families with children aged 7–14, which might be overestimated by the household dietary record method [10]. The present study using 24-h urine measurements showed small-to-moderate correlations increasing from 0.1 to 0.3 in the rank of sodium, potassium and sodium-to-potassium ratio.

On one hand, the familial correlations were likely to be attributable to similarity in dietary intakes of a family. For families with dietary intake mainly from home cooking, the food purchase and home cooking of the whole family was usually determined by one home chef, usually mothers in China, which echoes the relatively high associations of mothers among the family members found in this study. Besides, grandparents in China may live with their adult children to share their childcare burden and they could also play the role of home chef. As shown from this study, about 46.2% of the participating families had at least one grandparent living with the children. It suggests that family plays an important role in impacting the dietary intake of children, as parents or grandparents could shape children’s eating habits by selecting foods and preparing meals and snacks at home and also influencing food selections when eating out of home [19].

On the other hand, the small-to-moderate association may reflect the variety of dietary intakes among family members beyond home cooking. For example, family members could have various options for prepackaged food or takeaway food; schoolchildren may eat their lunch and choose extra snacks at school. It was reported the school meals contributed 33% of the sodium intake and 29% of the potassium intake for schoolchildren in Indonesia [20]. It was estimated in our study that about 60% of children ate their lunch at school during the weekdays. The ICC of schools also indicated the clustering of sodium and potassium intakes of schoolchildren within schools. As shown in the study of Ma et al., the social networks of children including their families, teachers and peers were all associated with their salt intake behaviors [21]. Apart from meals eaten out of home, the individual preferences for food and snacks could also explain the diet differences between children and the rest of the family.

In view of the wider factors beyond home environments, food environments including the availability of healthy food and food advertising aimed towards children are also important in influencing children’s dietary intakes and consequently, body weight. Hence, it is notable to integrate the sodium and potassium intervention action into the comprehensive strategies by engaging policy makers, schools, families and food industries to jointly promote healthy lifestyle behaviors and maintain healthy body weight in children, and this will bring lifelong benefits to the children.

The strengths of our study include: first, we collected 24-h urine, with 93% of participants collecting two complete 24-h urines and 7% one 24-h urine, which provided accurate assessment of sodium and potassium excretion. The bespoke electronic data capture system and stringent protocol ensured the high quality of the 24-h urine collection, with the complete rate of urine collections for 3552 person-days as high as 96.1%; second, we collected data from both a child and two adult family members in a family simultaneously, which provided a good opportunity to investigate familial aggregation of diet; and third, the sample size of 590 schoolchildren and 1180 adult families was large compared with other similar studies with 24-h urinary excretions, leading to relatively robust estimations of the sodium and potassium intakes.

The limitations of this study are as follows. First, this study did not collect the energy intake and dietary sources of sodium and potassium intake, the information of which may contribute to further explanation of the differences between children and adult family members. Second, we estimated the sodium and potassium intake directly from their urinary excretions, which could result in underestimations of intake as it has previously been reported that the urinary excretion of sodium and potassium represented 80–95% and 63–77% of their total intakes in adults as was the case in children [22,23,24]. However, the underestimation would not affect the urinary excretion collection recommended as the most accurate method for estimation of sodium and potassium intake among the various dietary intake estimation methods. Third, as the data used in this present study is of cross-sectional nature, no causal relationship can be inferred.

## 5. Conclusions

Children in China are consuming too much sodium and significantly inadequate potassium. The sodium, potassium excretion and sodium-to-potassium ratio of children are positively associated with their family excretions in small to moderate extent. Efforts are warranted to support initiatives of salt reduction and potassium enhancement in children through comprehensive strategies engaging with families, schools and food environments.

## Figures and Tables

**Figure 1 nutrients-13-02864-f001:**
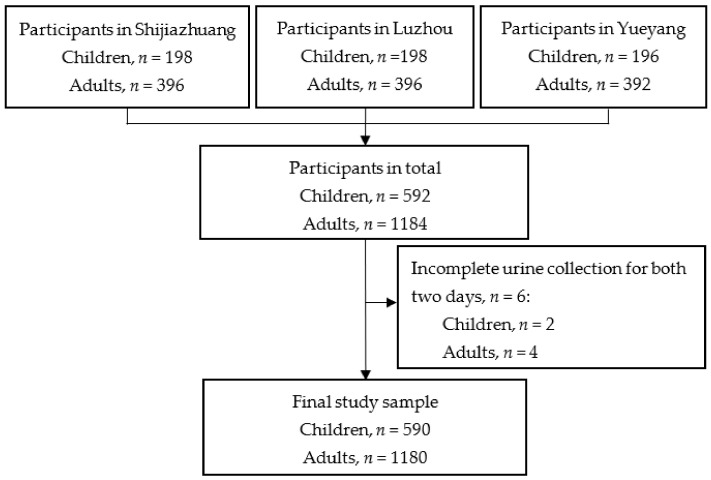
Flow chart of participants.

**Figure 2 nutrients-13-02864-f002:**
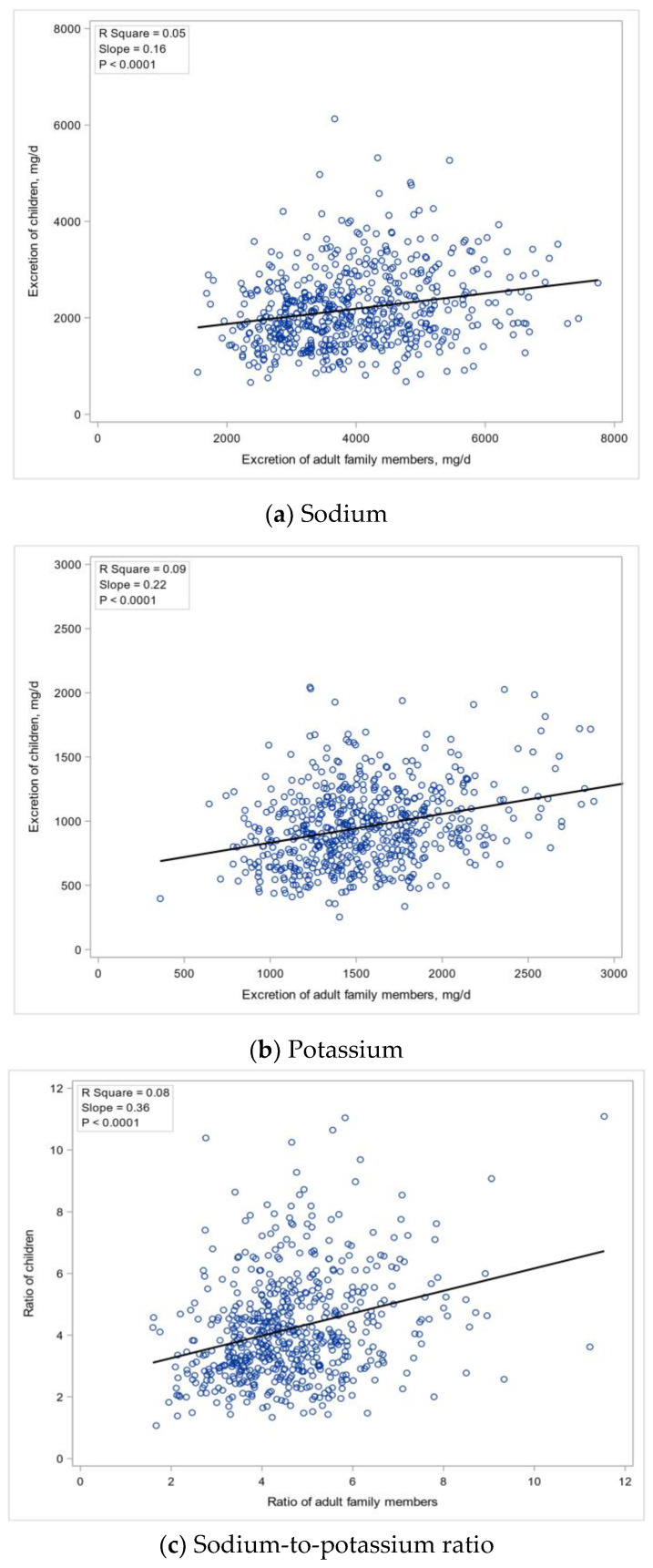
Distribution of sodium, potassium excretion and sodium-to-potassium molar ratio of the children and their adult family members. (**a**) Sodium, (**b**) Potassium, (**c**) Sodium-to-potassium ratio. The black line represents the regression line.

**Table 1 nutrients-13-02864-t001:** Characteristics of children and adult participants.

Indicators	Children	Adults
Total sample	590	1180
Area, *n* (%)		
Shijiazhuang	196 (33.2)	396 (33.6)
Yueyang	196 (33.2)	390 (33.0)
Luzhou	198 (33.6)	394 (33.4)
Male, *n* (%)	308 (52.2)	549 (46.5)
Age (y) ^1^	8.6 ± 0.4	45.8 ± 12.9
Body height (cm)	131.7 ± 6.6	161.2 ± 9.1
Body weight (kg)	30.2 ± 7.2	65.4 ± 12.6
BMI (kg/m^2^) ^1^	17.2 ± 3.0	25.1 ± 3.7
24-h sodium excretion (mg/d) ^1^	2180.9 ± 787.1	3948.6 ± 1406.8
Excess sodium, *n* (%)	455 (77.1)	1129 (95.7)
24-h potassium excretion (mg/d) ^1^	955.6 ± 310.1	1551.2 ± 528.2
Inadequate potassium, *n* (%)	590 (100.0)	1172 (99.3)
Sodium-to-potassium molar ratio ^1^	4.2 ± 1.7	4.6 ± 1.7
24-h urine volume (mL/d) ^1^	845.8 ± 297.7	1564.4 ± 588.7
24-h urinary creatinine (mmol/d) ^1^	4.9 ± 1.2	10.8 ± 3.1
Adult pairs in a family (*n*, %)		
Mother and father	318 (53.7)	
Grandmother and grandfather	137 (23.1)	
Other pairs	137 (23.1)	
Relationship with children (*n*, %)		
Mother		391 (33.1)
Father		375 (31.8)
Grandmother		233 (19.7)
Grandfather		165 (14.0)
Other adult relatives		16 (1.4)

Note: BMI, body mass index; ^1^ Mean ± SD.

**Table 2 nutrients-13-02864-t002:** Correlation of indicators between children and the average of their two adult family members.

Indicators	Pearson’s Correlation Coefficients	95% CI	*p*
Body height (cm)	0.507	0.445, 0.565	<0.0001
Body weight (kg)	0.385	0.314, 0.452	<0.0001
BMI (kg/m^2^)	0.304	0.229, 0.376	<0.0001
24-h urine volume (mL/d)	0.334	0.260, 0.404	<0.0001
24-h urinary creatinine (mmol/d)	0.255	0.178, 0.329	<0.0001
24-h sodium excretion (mg/d)	0.223	0.145, 0.299	<0.0001
24-h potassium excretion (mg/d)	0.305	0.229, 0.376	<0.0001
Sodium-to-potassium molar ratio	0.289	0.213, 0.361	<0.0001

**Table 3 nutrients-13-02864-t003:** Relationship of sodium, potassium excretion and sodium-to-potassium ratio of children with corresponding excretion of adult family members (*n* = 590).

Variables	Sodium Excretion of Children (mg/d)		Potassium Excretion of Children (mg/d)		Sodium to Potassium Molar Ratio of Children	
	β (95% CI)	*p*	β (95% CI)	*p*	β (95% CI)	*p*
Excretion of adults ^1^	0.11 (0.06, 0.16)	<0.0001	0.20 (0.15, 0.26)	<0.0001	0.36 (0.26, 0.45)	<0.0001
Age (year)	81.26 (−74.14, 236.65)	0.305	16.33 (−47.43, 80.09)	0.615	0.03 (−0.34, 0.39)	0.891
Sex (boys vs. girls)	403.09 (290.14, 516.05)	<0.0001	65.53 (18.73, 112.33)	0.006	0.56 (0.29, 0.82)	<0.0001
Body weight (kg)	32.13 (23.99, 40.28)	<0.0001	9.07 (5.75, 12.38)	<0.0001	0.02 (0.003, 0.04)	0.026

Note: ^1^ Mean sodium intake of the two adult family members (mg/d) for sodium intake of children (mg/d); mean potassium intake of two adult family members (mg/d) for potassium intake of children (mg/d); mean sodium-to-potassium molar ratio of two adult family members for sodium-to-potassium molar ratio of children. Mixed models were performed with city and school nested within city as random factors.

**Table 4 nutrients-13-02864-t004:** Association of sodium, potassium excretion and sodium-to-potassium ratio of children with different family members.

Family Relations	Sodium (mg/d)		Potassium (mg/d)		Sodium-to-Potassium Molar Ratio	
	β (95% CI)	*p*	β (95% CI)	*p*	β (95% CI)	*p*
Total children						
Mother (*n* = 390)	0.10 (0.05, 0.15)	0.0002	0.18 (0.12, 0.23)	<0.0001	0.30 (0.20, 0.41)	<0.0001
Father (*n* = 374)	0.04 (−0.01, 0.10)	0.112	0.19 (0.13, 0.25)	<0.0001	0.27 (0.17, 0.37)	<0.0001
Grandmother (*n* = 232)	0.10 (0.03, 0.17)	0.007	0.05 (−0.02, 0.12)	0.189	0.17 (0.05, 0.30)	0.007
Grandfather (*n* = 164)	0.09 (0.02, 0.16)	0.013	0.07 (−0.001, 0.14)	0.053	0.17 (0.03, 0.30)	0.015
Boys						
Mother (*n* = 203)	0.14 (0.07, 0.22)	0.0002	0.20 (0.12, 0.28)	<0.0001	0.32 (0.16, 0.49)	0.0001
Father (*n* = 186)	0.04 (−0.03, 0.12)	0.247	0.21 (0.13, 0.29)	<0.0001	0.32 (0.17, 0.48)	<0.0001
Grandmother (*n* = 125)	0.15 (0.04, 0.27)	0.009	0.08 (−0.03, 0.20)	0.149	0.13 (−0.06, 0.31)	0.170
Grandfather (*n* = 83)	0.10 (−0.01, 0.20)	0.071	0.07 (−0.05, 0.19)	0.267	0.17 (−0.03, 0.36)	0.087
Girls						
Mother (*n* = 187)	0.07 (−0.004, 0.14)	0.065	0.14 (0.06, 0.22)	0.0007	0.28 (0.16, 0.41)	<0.0001
Father (*n* = 178)	0.05 (−0.03, 0.12)	0.240	0.15 (0.07, 0.24)	0.0008	0.20 (0.08, 0.31)	0.001
Grandmother (*n* = 107)	0.05 (−0.04, 0.14)	0.266	0.02 (−0.07, 0.12)	0.632	0.28 (0.11, 0.45)	0.002
Grandfather (*n* = 81)	0.07 (−0.03, 0.17)	0.184	0.07 (−0.02, 0.16)	0.107	0.16 (−0.05, 0.36)	0.133

Note: Mixed models were performed with city and school nested within city as random factors and adjusted age, sex (in total-children models) and body weight of children.

## Data Availability

The data that support the findings of this study are available from the corresponding author upon reasonable request after the publication of major results of the trial and in compliance with the pertinent regulations of data management and data sharing in China.

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
