# Peer review of "Sodium and Potassium Excretion of Schoolchildren and Relationship with Their Family Excretion in China"

_nutrients, 2021, doi:10.3390/nu13082864_

Round 1

Reviewer 1 Report

All comments below are also uploaded in a word-document.

The beneficial and harmful effects of sodium and potassium intake has been and still is an interesting and influential subject. In fact, it has become more and more relevant given the fact that an increased intake of processed foods and increasing numbers of individuals with obesity among adults and even children is seen worldwide. Therefore, providing reference on the subjects as was done in the current study can be very useful. Overall the paper was well written and much of it well described. The authors provide useful information that could also benefit future intervention studies. However, a lot of valuable information, that might even be readily available to the authors, could receive more attention in the article. Obviously, mixed models statistics and providing correlations does not provide information on causal mechanisms, but providing more information on energy intake, source of intake and household structure/habits would greatly help to provide more perspective. 

Major comments:

  1. Overall, the authors have done a good job at providing reference for sodium and potassium intake aggregated in families in China. However, it would be useful for readers to receive more information on household structures, as was done for schools and cities. It would be very interesting, for example, to know whether grandparents are part of the same households, given the fact that in China grandparents often lean in with childcare. This would greatly influence the strength of the correlations between sodium and potassium intakes of children, parents and grandparents.
  2. Furthermore, the authors should provide more data on energy intake. In fact, on page 5, lines 175-177, the authors mention that they performed a ‘energy corrected sensitivity analysis’. Currently, in the text, it is not clear what the authors have done and does not provide any reproducibility, especially given the fact that they do not corroborate what variables are incorporated in this ‘energy’ variable . From the baseline characteristics it is also not clear what variables are probably used for this analysis. From supplementary tables 1-3, one could only conclude that this energy correction is solely based on literature, and not on own data. The authors should explain this additional analysis in more detail.
  3. At the end of the article, (Page 8, lines 222-221), the authors claim that the results found significant familial correlations between sodium, potassium intake and sodium-to-potassium ratio of children and their adult family members. This statement is too bold. Especially given the fact that correlations between children and grandparents (especially with grandfathers) are mostly non-significant. I suggest the authors rewrite this section of the discussion and corroborate on the matter.

Minor comments

  1. Page 2, lines 75-76. Subjects were selected in the order of grandparents, parents, uncles and aunties. Intuitively, it would be expected that children have greater correlations with their parents. It is an interesting choice to prioritize grandparents. Nor in the reference, nor in the discussion this choice is further corroborated. I would suggest the authors to do so.
  2. Page 2, lines 80-81 Two consecutive 24-hour urine samples were collected. The authors do not mention the time in between collections. Is this the same for every subject, otherwise a median time in between collection would be interesting for readers. Also, given the fact that these children eventually participated in a controlled trial where a sodium intervention was used, it would be interesting to know whether 24-hour excretions were all collected before the trial had started.
  3. Page 6, lines 181-186 (figure 2). The authors should add additional statistical information. R-square or percentage explained would be of interest.
  4. Page 7, lines 187-195. ‘Intake of adults’, as stated in table could be misleading, changing it to ‘intake’, with the description underlined, would be sufficient. Moreover, in table 2, the authors present the results in too many digits. The authors should restrict to an appropriate number of digits. This also accounts for table 3 and supplementary tables 2-3.
  5. Page 8, line 203. These age group, should be this age group (singular instead of plural).
  6. Page 8, lines 204-206. It would be interesting for readers to also mention the % of above recommended intake for sodium and potassium for adults, as is done for children.
  7. Page 8, lines 222-221. This statement is too bold. Especially given the fact that correlations between children and grandparents (especially with grandfathers) are mostly non-significant. Also see major comments.
  8. Page 8, lines 233-250. It seems that the background information is based on literature in this section. It would be interesting to also use available information from the database of the authors. Is there any information on dietary intake and the source of this dietary intake?
  9. Page 9, line 274. ‘and so as’ should be ‘and so was’ or ‘as was’.

Reviewer 2 Report

Comments in attached document. Abstract, Introduction and Discussion section require extensive English editing.  

Apart from school meals, what opportunity fo the children have for a diet that differs from that of the rest of the family? For example, can they choose different foods from the adults. Is there a salt cellar available at table? These questions regarding Chinese habits can clarify the nature of the correlations for international readers unfamiliar with the eating environment of the studied sample.

Reviewer 3 Report

This is an interesting study that examined the relationship between children's and parent/adult family member's sodium and potassium intake, which were assessed using two 24-hour urine collections. It is certainly important to explore this relationship to better understand the influences on children's sodium and potassium intake, which could affect future cardiovascular disease risk. This could have implications for designing dietary interventions and public health programs for both children and adults/families. While this paper has many merits, it requires extensive editing and possibly some reanalysis before it should be considered for publication. 

There are a number of sentences that are difficult to read, are awkwardly worded, or have English language errors in them. I would recommend having a native English speaker look over this, if you haven't already.

In addition, it is important to remember that you measured 24-hour urinary sodium and potassium excretion. These are great biomarkers of intake, and certainly the gold standard method for estimating intake, but remember it is not a direct measurement of intake. Avoid using the word "intake" when presenting your results, and instead use "excretion" or "24-hour urinary excretion" to be more accurate. I have pointed out a few of these instances in my specific comments below, but I could not list every instance. However, it is appropriate to refer to "intake" in the intro and discussion when you are contextualizing your study (i.e. to remind the reader why you measured 24-hour sodium and potassium excretion). 

Introduction:

  • Lines 51-52: I'm not sure I understand your point here. Are you saying that because multiple 24-hour urine collections are burdensome, this measurement has not been done in children? 
  • Lines 54-57: So previous studies have looked at the relationship between parent and child sodium and potassium intake. Is your point in this paragraph that your study is novel because you are using a better method for estimating sodium and potassium intake? If so, I think this entire paragraph needs to be rearranged to make that more clear. Start with the information in lines 54-57, then mention that a weakness of these studies is that they did not use the gold standard method for estimating sodium and potassium intake, then include the information in lines 49-52 about why they likely did not collect these measurements, then introduce the aim of your manuscript. 

Methods:

  • Lines 72-73: Why did you require that the families eat homemade meals at least 4 times per week? This seems like it would create a very biased sample of families. In addition, if families are frequently eating the same meals together, then of course you'd expect sodium and potassium intake to be correlated among members. 
  • Line 76: "aunties" is more of a slang/informal term for "aunts." In this context "aunts" would be better.
  • Lines 80-92: Much more detail is needed about how and when all of these measurements were collected. How were sodium, potassium, and creatinine measured in the urine? What instrument(s) did you use? Were the samples from children and adults within the same family collected on the same day? How was physical activity measured? When were height and weight measurements taken, and with what tools? Were they taken in a laboratory setting? These details may be included in other referenced publications, but they need to be included in the methods section of this manuscript as well. 
  • Lines 97-98: You mention in the introduction (lines 49-51) that multiple 24-hour urine collections are more accurate, and imply that the lack of multiple 24-hour urine collections in children is a limitation of previous studies. If that is the case, why did you determine it was acceptable to only use one day of urine for some participants, rather than excluding them entirely? As you stated, one of the strengths of your paper is the use of two 24-hour urine collections, so I would recommend excluding any participants that do not have two complete urine samples. 
  • Lines 101-102: do you mean scatter plots?
  • Lines 100, 104, 113: I know that urinary sodium and potassium excretion are indicators of intake, but you should avoid using the word "intake" and instead say "urinary sodium excretion." Even though it's a biomarker of intake, it's not a direct measurement of intake.
  • Line 106: I think this is supposed to read "...with family and school nested within city..." Also, what about class? Wasn't that another nested variable?
  • Lines 117-121: Do these estimated energy needs take into account height and weight? Are you assuming that participants were consuming exactly the number of calories that they need, based on these numbers? Do you know if they were weight stable? I have doubts about this strategy for correcting for energy intake unless you can point to a reference where this has been done before. I also don't think it's necessary. This manuscript seems to be focused more on the  correlations between sodium and potassium excretion in adults and children. If you were directly comparing absolute sodium and potassium values between adults and children, for example, then yes it would be important to correct for energy intake. 

Results:

  • Table 1: Why do you have both sodium and salt intake? For both sodium and potassium, if those are the results of the urinary excretion measurements, then labeling them as intake is incorrect. They should be labeled as 24-hour urinary sodium excretion and 24-hour urinary potassium excretion in the left column. 
  • Table 1: The correlations between children and adults should be reported in a separate table, with r-values reported for all correlations run (not just the ones that were significant) and exact p-values and/or confidence intervals reported.
  • Lines 161-179: again, you're not actually reporting intake, you're reporting urinary sodium excretion. The text should discuss "excretion" and not "intake." 
  • Figure 2 and Table 2: excretion, not intake
  • Lines 170-175: You should include your statistics in the text (p-values and possibly beta coefficients as well). 
  • Lines 170-172: Unless I'm misunderstanding, this is not true for sodium based on the data presented in Table 3.
  • Lines 258-259: But you also included participants who only had one valid 24-hour urine collection in your analyses.

Reviewer 4 Report

Li and colleagues proposed a cross-sectional study of 592 families comprising 590 children and 1180 adults to evaluate sodium and potassium concentrations in two consecutive 24-hour urine collections. Following the evaluation of the mean levels, the authors evaluated the family correlation by Pearson's coefficient between indicators in children and adults. Subsequently, the association of sodium or potassium excretion between children and family members was evaluated. In the adjusted mixed models, various associations have been highlighted.
Although the work is interesting, the authors use the concept of urinary excretion and sodium and potassium intake indifferently. This is incorrect and can create misunderstandings, also considering that the sodium and potassium intake can mutually influence their respective levels in the urine. In addition, the recommended intake levels were compared with urine levels. I believe this approach is imprecise and should be evaluated with caution.
I advise the authors to standardize the wordings, limiting to referring only to renal excretion and not to the intake when the results are presented. Although urine levels are directly related to intakes, this correlation can be influenced by various factors and the authors did not evaluate the participants' actual intakes. Carry out the same standardization also in the title and the tables.
At lines 161-167 in the data description of Table 2, I recommend reviewing the correctness of the values ​​and consistency with the tables.
Minor aspects:
Review the formatting of the references along with the text (bibliographic reference numbers should be inserted between square brackets and not round).

Round 2

Reviewer 3 Report

All comments were thoroughly and sufficiently addressed. 

Reviewer 4 Report

no other comments